# Fertility Preservation Options for Transgender Patients: An Overview

Natalie Mainland [1], Dana A. Ohl [2], Ahmed R. Assaly [1], Nabila Azeem [1], Amber Cooper [3], Angie Beltsos [3], Puneet Sindhwani [4] and Tariq A. Shah [4],*

1   College of Medicine and Life Sciences, University of Toledo, Toledo, OH 43606, USA
2   Department of Urology, University of Michigan, Ann Arbor, MI 41809, USA
3   Kindbody, St. Louis, MO 63141, USA
4   Department of Urology, University of Toledo, Toledo, OH 43606, USA
*   Correspondence: tariq.shah@utoledo.edu

**Abstract:** Fertility preservation technologies have existed for decades, and the field is rapidly advancing; limited data exist regarding the use of these technologies by transgender patients. Many options are available for transgender patients who wish to preserve fertility before transitioning. These options include the cryopreservation of gametes, embryos, or ovarian tissue. Currently, ejaculated, or testicular sperm, immature oocytes, and ovarian tissue can be preserved for later use, but no such use option exists for immature testicular tissue. Many financial, sociological, and legal barriers and a lack of awareness among physicians and patients also hinders the utilization of these fertility preservation services. While options are abundant, usage rates are relatively low. The initial data regarding the successful use of preserved tissues appears promising, with birth rates not dissimilar to non-transgender patients. Further investigations into this area are needed. In addition, counseling regarding fertility preservation options should become a significant part of the provider-patient conversation before transitioning therapies.

**Keywords:** fertility; transgender; reproductive technology





## 1. Introduction

Over the last two decades, advances in reproductive technology (ART), such as in vitro fertilization, and cryopreservation technology have given rise to newer areas of reproductive tissue banking such as the freezing of sperm, eggs, embryos, testicular tissue, strips of intact ovarian tissue, or even entire ovary [1]. Cryopreservation of sperm, oocytes, and embryos is now routine in infertility management, while the other above-mentioned techniques are still in their early stages of development.

Though cryopreservation is routine in managing chemo or radiation-therapy-induced infertility and specific non-cancer indications (e.g., advanced female age), a new need has more recently emerged: patients who plan medical gender transition. This transition process commonly includes surgical removal of reproductive organs as well as hormone therapy, with resulting infertility. Recent studies have shown that many transgender patients desire to have children, and the success of cryopreservation in other circumstances suggests that it is also a viable option for preserving fertility in transgender patients.

Although fertility preservation presents an exciting opportunity for transgender patients, there are potential problems. Lack of awareness is one major issue. Recommendations exist from organizations such as the Endocrine Society to counsel transgender people about the possibility of preserving fertility prior to transition, but this counseling is often inconsistently performed [2,3]. This mirrors past experience with cancer patients, where counseling about fertility preservation has historically been underutilized [4]. In addition, sociological and psychological factors play a role, with social determinants of health having a profound impact on perceptions of parenthood [5–7]. Should the above be overcome,

economic barriers still remain, such as the expense of the preservation procedure often deemed "elective" and storage fees for preserved material, which are often not covered by insurance in the United States.

Finally, as with any medical process, ethical and legal factors must be taken into consideration. As transgender care for youth becomes more commonplace, patients who would not yet otherwise be thinking about reproduction must confront choices that profoundly shape their futures, again mirroring the experience of young cancer patients. Legal problems also exist, including consent/assent concerns, in a system that has not yet adapted to new possibilities.

## 2. Fertility in Transgender Patients

Patients undergoing medical transition have a high rate of fertility loss due to their treatments during the transition. The most common gender transition therapies among male-to-female ('male at birth') individuals include feminizing estrogen hormone therapy, breast enhancement, chondrolaryngoplasty reduction, penectomy, orchiectomy, and vaginoplasty. For transgender men ('female at birth'), treatments include masculinizing testosterone hormone therapy, bilateral mastectomy, and hysterectomy/oophorectomy. While surgical procedures such as hysterectomy/oophorectomy or orchiectomy obviously preclude conception due to loss of gamete production, pharmacological treatments such as cross-sex hormone therapy also have an impact. For patients on estrogen therapy who do not undergo orchiectomy, estrogen may have permanent deleterious effects on sperm quality [8]. For patients on testosterone therapy without hysterectomy/oophorectomy, there does not appear to be a permanent impact on fertility; indeed, many patients who are actively on testosterone can still become pregnant. However, testosterone is a known teratogen and cannot be continued during pregnancy [9].

Many transgender individuals are of reproductive age at the time of gender transition. While parenthood is generally not an immediate priority at this time, studies have indicated a strong desire among transgender persons to be parents at some point in their lives. A 2012 Dutch study found more than 54% of transgender women who had undergone surgical transition expressed a current or previous desire to procreate [10]. This would entail possibly delaying hormone therapy to allow sperm cryopreservation before transition, or utilizing donor sperm banks. However, another study in 2012 found that 37.5% of transgender men reported that they would have considered freezing reproductive tissue had the option been available, and 54% of them expressed a desire for children. In the US, a 2016 study of transgender men in San Francisco found that 15% reported a desire to become pregnant [11].

## 3. Cryobiology of Gonadal Tissue Freezing

Thanks to advances in ART, the loss of fertility consequent to gender reassignment therapies need not be inevitable. Rigorous pre-procedure counseling followed by preservation of reproductive cells or tissue is a clinically viable alternative.

The cryopreservation of cells and biological tissue is a process that entails brief exposure to high molar concentrations of a mixture of a variety of cryoprotectant agents (glycerol, propanediol, dimethyl sulfoxide, or sucrose), which leads to complete dehydration of the cell contents to prevent ice crystal formation that may damage the cells, followed by vitrification of specimens at extremely low temperatures ($-196$ °C) in liquid nitrogen. In vitrification, a solution transitions from a liquid state to a vitreous state at low temperatures, circumventing the formation of ice crystals to prevent tissue damage—a significant advantage over previously-common slow freezing.

Cryopreservation is an effective way to preserve fertility and the use of preserved tissue frequently results in a successful pregnancy [12,13]. However, the success of the preservation process depends greatly on the quality of the material to be preserved. There is a narrow window of only a few years to undergo preservation of reproductive tissue,

especially for eggs, embryos, or ovarian tissue. The older the patient, the harder it becomes to achieve current or future parenthood even through highly successful ART [14].

### 3.1. Sperm Freezing

Frozen sperm obtained via masturbation or through testicular or epididymal needle aspiration has been successfully utilized for procreative purposes for decades [15]. While the characteristics of transgender women's semen specimens may not be optimal [16], there is evidence that such samples can, nevertheless, be used to initiate conception [17,18]. Transgender individuals face unique challenges with sperm quality: transgender women taking hormone therapy have been observed to have significantly altered sperm parameters, which may persist for months after treatment discontinuation [19]. Even individuals not taking hormones may experience difficulty: genital tucking and/or wearing tight undergarments is common among transgender women, and these practices negatively impact sperm quality due to increased heat stress and greater production of reactive oxygen species [20,21]. The sperm abnormalities in these individuals range from low count, low motility, and low normal forms, to azoospermia, in which case fine needle aspiration from the epididymis or the testis may be necessary [22].

Cryopreservation of sperm inevitably leads to a reduction in the usable quality and quantity of sperm at the time of thawing. While this scenario is not any different from that of non-transgender fertility patients, it may nevertheless constitute a major setback to many transgender patients; suboptimal samples inevitably require the use of expensive assisted reproductive techniques such as in vitro fertilization (IVF) or IVF with intracytoplasmic sperm injection (ICSI). These techniques, collectively known as ART, tend to be quite expensive and insurance usually does not cover them. Only an insignificant minority is likely to conceive without ART, using less expensive methods such as intrauterine sperm insemination. Even cryopreserved sperm from optimal samples show significant increases in abnormal morphology and decreases in motility after thawing; current research focuses on altering the cryoprotectant medium to prevent damage [23].

Currently, limited information is available regarding pregnancy rates using cryopreserved sperm collected from transgender patients prior to transition. However, other patient populations have long had success with the use of cryopreserved sperm, with fertilization rates not significantly different from those of fresh sperm even for patients with suboptimal semen samples; e.g., patients with azoospermia [24–26].

As for prepubertal patients, the preservation and use of testis tissue from such patients is not yet feasible. The preservation of fertility in prepuberty is also of particular relevance to childhood cancer patients, and this is a topic of research [27]. Currently, no standardized protocols exist for the preservation of this tissue, and techniques vary with regard to cryoprotectant medium and freezing rate [28]. Protocols for the preservation of mature testicular tissue are not applicable; these focus on preserving spermatozoa, with no consideration for preserving spermatogonial stem cells, which are needed for spermatogenesis [28]. The best way to utilize such tissue, whether by grafting or in vitro spermatogenesis/maturation (IVM), is not known. Of these two techniques, tissue grafting has shown success in non-human animals, including primates, although ICSI was still required for fertilization [28]. IVM has been successful with rodent tissue, but in humans, it has only reached the spermatid stage [28].

As of 2022, some fertility centers have elected to cryopreserve testicular tissue from prepubertal patients in anticipation of technological advancements that would facilitate its use [29,30]. However, to our knowledge, no live births have yet occurred using cryopreserved tissue harvested from prepubertal individuals [28].

### 3.2. Oocyte and Ovarian Tissue Freezing

Embryo-freezing technologies have existed for more than three decades, while technologies for cryopreservation of oocytes have advanced dramatically in the last decade after the switch from slow-freezing to vitrification as the preferred cryopreservation method.

Oocyte cryopreservation has now become part of standard medical practice [31]. When desired, the oocytes can be thawed and fertilized with the sperm from the patient's partner or a sperm donor. Current data show that the pregnancy rates achieved after uterine transfer of embryos derived from cryopreserved oocytes have paralleled the success rates achieved with the use of fresh non-cryopreserved oocytes and embryos [32] and the overall projected success rates based on extrapolations of current data are quite high [33]. However, some studies have shown that although pregnancy rates may be similar, the successful creation of an embryo is likely to require more thawed oocytes than fresh oocytes, and the number of oocytes required for successful outcomes increases with maternal age [34,35].

Among transgender patients, the use of cryopreserved oocytes as opposed to embryos is similarly novel. In 2013, in the first report of its kind among transgender people, a biological female, self-identified as male, underwent an oocyte cryopreservation procedure prior to undergoing gender transition [36]. Since then, this practice has become more common, but overall rates of oocyte preservation among transgender men remain low. [37–39]. Since testosterone generally induces an anovulatory state in biological females, it has been hypothesized that transgender men may experience issues with oocyte harvesting for cryopreservation. However, a 2023 study compared results of oocyte vitrification in testosterone-naïve patients vs. patients who had previously used testosterone, but had ceased treatment at least three months prior to oocyte harvesting, and found no difference in outcomes between the patient groups [37].

While the use of cryopreserved oocytes in transgender patients specifically remains low enough that overall rates of successful pregnancy cannot be extrapolated, the delivery of normal healthy infants has been reported in multiple cases [17,39]. This mirrors the experience of non-transgender patients, where the use of cryopreserved oocytes is also quite low, with the majority of patients not returning to use their oocytes; indeed, among patients who cryopreserve oocytes in anticipation of iatrogenic infertility (e.g., gonadotoxic chemotherapy), the utilization rate is under 10% [34].

Ovarian tissue cryobanking has only recently begun the move from an experimental procedure into clinical practice. Clinical and laboratory protocols for optimization of various steps involved in ovarian tissue cryobanking, i.e., obtaining ideal grafts, freeze-thaw protocols, transplantation, and successful regrafting, are the focus of vigorous research at fertility clinics worldwide [31]. Ovarian tissue freezing often entails the excision and freezing of several strips of ovarian cortical tissues [40]. The basis for this concept lies in the preservation of tissue in cancer patients undergoing treatment. The cortical strips can be re-transplanted on the same ovary when the patient is ready to resume reproductive processes [41]. Alternatively, the cortical strips can also be implanted heterotopically in the patient's forearm [42]. Globally, the rates of successful live birth using preserved ovarian tissues are about 33% [43]. A comprehensive analysis of patients implanted with previously cryopreserved ovarian tissue at reproductive medicine centers throughout Germany, Austria, and Switzerland from 2007–2020 found ovarian tissue transplantation to yield pregnancy and live-birth rates of 32.7% and 26.5%, respectively, though these rates varied depending on the size and experience of the medical center [44].

Cryopreservation of ovarian cortical strips (or, potentially, the whole ovary) opens new opportunities. It is the only fertility preservation option suitable for prepubertal patients, as ovarian stimulation, oocyte harvesting, and cryopreservation are not feasible to perform in this age group [45]. It also has the advantage of being faster for the patient; there is no need for patients to potentially undergo multiple cycles of ovarian stimulation and egg retrieval [46].

This may be particularly useful for prepubertal transgender patients or patients concerned about delaying hormonal therapy, as it allows for fertility preservation without requiring the production of mature oocytes. However, pregnancy using these tissues requires discontinuation of testosterone therapy and may not be well-tolerated.

This method is not without drawbacks; post-harvest graft ischemia, resumption of hormonal function and follicular development after transplantation, and in vitro matu-

ration of oocytes continue to pose challenges [47]. Furthermore, there currently is some question as to whether vitrification (now the preferred cryopreservation method for sperm, mature oocytes, and embryos) [48] is truly superior to slow-freezing for ovarian tissue. Vitrification involves higher concentrations of cryoprotectants than slow-freezing, which can be toxic to cells and primordial follicles [49]. A 2021 meta-analysis found vitrification to preliminarily be superior, but noted that high variability in protocols and a lack of data on clinical outcomes suggest that further research is required before a definitive conclusion can be drawn [49].

As for immature oocytes, in a case report from 2020, physicians in Tel Aviv reported successfully harvesting and cryopreserving immature oocytes from a prepubertal girl with mosaic Turner syndrome [50], but cryopreserving oocytes from prepubertal patients is not yet common practice, and to date, there are no records of any pregnancies using oocytes harvested from such patients. These areas are the focus of intense investigation [51].

### 3.3. Freezing of Embryos

In addition to the freezing of gonadal tissue, there is the option to cryopreserve embryos, which has been performed successfully for decades [31]. Embryos can be created in the laboratory using the transgender individual's own oocytes/sperm, or oocytes/sperm from either a partner or a third-party donor. Embryos created by IVF can be frozen, and in the future transferred to the uterus of the female partner or a surrogate at a convenient time [52]. Children born as a result of pregnancy with vitrified embryos show no difference in growth, development, or overall health compared with children born of pregnancy with fresh embryos [53]. However, the cryopreservation of embryos is more legally and ethically complex than that of gonadal tissue; as the embryos contain genetic material from two people, both may have shared ownership and the ability to prohibit the other party from using the embryos to conceive [33]. Surrogacy also comes with issues; finding a surrogate mother can be difficult and expensive, and although surrogate mothers overall report the experience as fulfilling, it is still often fraught with emotional turmoil for both the surrogate and the couple [54]. Furthermore, surrogacy is not legal everywhere: it is prohibited in some countries and some US states, and even in places where surrogacy is legal, a judicial determination of parentage may be necessary to secure parental rights [55].

Table 1 summarizes current feasible options for fertility cryopreservation, their advantages, and disadvantages.

**Table 1.** Summary of preservation options, their advantages, and disadvantages.

| Patient | Cryopreservation Option | Advantages | Disadvantages |
|---|---|---|---|
| **Male at birth** | Freezing sperm | Only feasible option for male-at-birth patients to preserve sperm<br>High rate of success<br>Relatively inexpensive compared to other cryopreservation options | Cryopreservation may reduce sperm quality, necessitating expensive ART<br>Patient may have difficulty producing a sample<br>Not feasible for prepubertal patients |
| **Female at birth** | Freezing oocytes | Current standard practice<br><br>High rate of success | Only mature oocytes can be harvested<br>Requires ovarian stimulation, usually multiple times, which can be time-consuming and difficult for the patient |
| | Freezing ovarian tissue | Viable for prepubertal patients<br>Faster than oocyte harvesting; does not require cycles of ovarian stimulation | Tissue retrieval requires laparoscopic surgery<br><br>Lower success rate than oocyte freezing<br><br>Still novel, with optimal techniques not yet standardized; difficulties persist with in-vitro oocyte maturation and ovarian graft failure |
| **Both** | Freezing embryos | May have higher live birth rates compared to oocyte freezing<br><br>Tried-and-true; has been in use for decades | Both partners share ownership, creating possible legal issues regarding use, storage, and disposition<br>Potential difficulty finding a surrogate to carry pregnancy<br>Requires both sperm and oocyte, necessitating either a partner or a donor |

## 4. Trends in Demand and Utilization

### 4.1. Demand for Fertility Preservation among Transgender Patients

Although guidelines from WPATH, the Endocrine Society, and the American Society for Reproductive Medicine recommend counseling about fertility preservation prior to initiating hormone therapy [39], utilization of preservation services remains low. A study conducted at a Dutch fertility center in 2011 saw 15% of adult transgender women freeze sperm after receiving counseling about fertility preservation [10]. The desire for fertility preservation appears to be lower among transgender youth and young adults than among older adults. A 2016 study found that in post-pubertal transgender youth and young adults who had received counseling about fertility preservation, the utilization rate of fertility preservation services was 2.8% [56]. The most common reasons given for declining to preserve fertility included intention to adopt (45.2%) and desire to never have children (21.9%), while only 8% reported declining due to financial concerns. 1.4% reported declining due to discomfort with producing a semen sample, and a further 1.4% declined because they did not wish to delay the start of hormone therapy [56]. A later study among a similar cohort yielded a utilization rate of approximately 5% [57], though, in that study, patient attitudes toward reproduction were not examined.

It is often assumed that young people are less likely to consider fertility preservation as they are not yet at an age where one typically thinks about starting a family [58]. However, transgender young adults are less likely to utilize fertility preservation than their non-transgender counterparts. In a study involving adolescent cancer patients at risk of infertility, 28.1% elected to bank sperm when counseled about their fertility preservation options [59]. Possible explanations for these disparities include economic barriers or psychological factors such as concerns about treatment or reproduction leading to worsening dysphoria [58].

### 4.2. Utilization of Preserved Tissues

While rates of fertility preservation among transgender patients are low, the rate of utilization of preserved tissue is even lower. Information regarding this is currently confined to case reports and/or small case series. In a case series spanning the years 2009–2021, of 44 adolescent/young adult transmen who presented to NYU for fertility preservation services, none returned to use their cryopreserved oocytes [38]. Very limited data currently exists regarding pregnancies derived from transgender patients' cryopreserved tissue, although a 2017 report described two transgender men who successfully used their preserved oocytes to conceive children, with the embryos being implanted in their female partners [17].

In spite of the low rates of utilization of both fertility preservation itself and the preserved tissues, it remains important to counsel patients about their options. A 2021 study of transgender individuals across a wide age range found that counseling about fertility preservation prior to transitioning was associated with lower levels of decisional regret [60]. Young patients, especially, may express regrets over not preserving fertility as they age [61] and become involved in stable, long-term relationships [58].

## 5. Barriers to Access

Although a lack of interest in reproduction has been cited as a factor in not pursuing fertility preservation [56], it is not the sole reason for low utilization rates. Indeed, for transgender patients, there are numerous barriers to access that have a profound impact on the perception and utilization of fertility preservation therapies.

### 5.1. Sociological Barriers

Economic disparities, generalized discrimination, and lack of insurance coverage among those in the transgender community all impact access to medical care for gender transition [62]. While presently, discrimination against LGBTQ persons lingers on as a significant global human rights challenge, recent years have witnessed a paradigm shift in

how different societies worldwide view the human rights of transgender and non-gender-conforming individuals [63].

There is a growing global recognition of the fact that social determinants of health (SDOH) affect physical and mental health outcomes [5]. SDOH are social factors that are essential to the preservation and optimization of health, prevention of disease, and guarantee of maximum quality of life. The LGBT community, in general, and transgender people in particular face group-specific barriers to care that impact their quality of life [64]. Gender affirmation remains a significant social determinant of the health and wellbeing of transgender individuals [6]. Over the years, four commonly accepted domains of gender affirmation—social, psychological, medical, and legal—have received much attention. Until recently, parenting as a gender affirmation domain has not been among the primary medical or psychological concerns for the transgender community. Rather, poverty, stigma, ridicule, and social rejection have remained their chief obstacles [65]. Conflicting global opinions and social norms regarding what constitutes marriage, fatherhood, motherhood, and family pose a significant challenge to the transgender community's aspiration for parenting [7,66]. Parenting, for many transgender individuals, has remained a scarcely-expressed aspiration or unfulfilled desire.

Reproductive rights and options for transgender patients are evolving [2,67]. The American Society for Reproductive Medicine has opined that denial of access to fertility services to transgender people is unjustified [68]. It emphasizes that current data do not support that children raised by transgender individuals are vulnerable to distinctive harm [68]. While conversations about unique risks and the lack of data about long-term outcomes must occur prior to initiation of treatment, requests for ART must be considered without regard for gender identity.

Advances in ART have opened both new opportunities and new challenges for parenting by transgender people, adding to a myriad of cultural, ethical, financial, religious, and psychological concerns surrounding their human and procreative rights. While many transgender patients would consider fertility preservation options if offered, it is not uncommon for them to be conflicted about their future fertility options; some desire to seek a clear break with their natal gender [69]. Nevertheless, the option of fertility preservation provides a means to avoid potential decisional regret [58].

*5.2. Financial Barriers*

Cost is a significant factor in fertility preservation. A 2016 study found the average cost for oocyte freezing in the United States to be USD 9253, while the cost for cryopreservation of a sperm sample retrieved by masturbatory emission averaged USD 745 [70]. These estimated costs include 1 year of storage, with additional storage costing an average of USD 343/year [70]. Furthermore, a 2020 study found price transparency to be an issue. Of the facilities listing sperm cryopreservation as an available service, only 17.7% had pricing information readily available [71]. Should patients use their cryopreserved tissue to conceive, the cost increases. A 2020 study of women who underwent cryopreservation prior to gonadotoxic chemotherapy found that when the cost of thaw and fertilization cycles and frozen embryo transfer was included, the total cost of a successful pregnancy with cryopreserved oocytes averaged USD 16,588 [72].

The issue of cost is compounded by the usual lack of insurance coverage for fertility services. Even for patients with health insurance, fertility preservation services frequently are not covered [73]. Some US states mandate coverage for fertility preservation, but even among these states the services covered and rates of coverage vary widely, and the mandate may not apply to all insurers [73,74]. Additionally, transgender patients are more likely than other patients to lack insurance coverage [75] and to earn income at a level below the poverty line [29] creating further barriers to care.

In countries where infertility care is covered by a national health care system, it is tempting to consider such services 'free.' However, the system bears the burden of any new procedure added to its covered services, leading to increased system costs. For example,

burgeoning costs in Nordic national healthcare systems have led to changes in healthcare delivery and limitations placed on what fertility services may be covered in a lifetime, and who may receive such services. In Finland, the conflict between cost-cutting at the national level and the perceived right of patients to unlimited care caused the Finnish government to resign in 2019. Even in systems of universal healthcare, fertility preservation of transgender patients is costly and may be limited by the system in order to reduce costs.

## 6. Legal and Ethical Dilemmas

Realistic expectations for fertility preservation therapy and all current and potential legal issues therewith must be clarified before starting any medical transition regimen. Patients must be prepared in the event the treatments do not bring about expected psychological or biological outcomes. For example, patients should be ready to make a plan for what will happen to the frozen reproductive tissue if the desire for future parenthood ceases to be a paramount concern, due to unanticipated circumstances.

### 6.1. The Importance of Informed Consent

Patients whose gametes, embryos, ovarian or testicular tissue is to be frozen must give their informed consent before the cryopreservation. The elements of an appropriate consent form should include the following:

(1)   A description of the procedures involved in freezing and thawing.
(2)   Procedural risks (failure of the tissue to survive freezing and thawing, and the theoretic risk of increased congenital anomalies; a mechanical failure of a catastrophic event leading to the loss of the frozen tissue, etc.).
(3)   The benefits (preservation of fertility).
(4)   Alternatives to cryopreservation (use of fresh donor tissue when available).

Furthermore, disposition of the frozen tissue in the case of the lack of continued wish for procreation [10], death or divorce or dissolution of a partnership, nonpayment of storage fees, and loss of contact with the person(s) should be covered in the consent form [76–78].

Finally, since many of the procedures for the use of tissue (such as prepubertal ovary or testis) are not yet perfected, strong consideration must be given to performing these procedures under a research protocol with Ethical Board approval.

### 6.2. Legality and Inheritance for Nonstandard Familial Relationships

The convergence of technologies related to cryopreservation of ejaculated sperm, testicular sperm, oocytes, embryo, or ovarian tissue with those of in vitro fertilization and related procedures has opened exciting new opportunities for men and women seeking medical gender transition. However, many technical, clinical, and ethical questions have yet to be addressed. There is no denying that issues related to cryopreservation of reproductive cells or tissues among transgender people have the potential to beget unprecedented ethical, financial, religious, and legal challenges, such as establishing biological and familial relationships on legal documents that cannot recognize fathers who gave birth or mothers who donated sperm. Tissue inheritance rights for those who do not end up using their preserved tissues further complicate these scenarios [79]. The answers to these questions await the results of additional research and discussion [68].

## 7. Conclusions

In the last two decades, options for fertility preservation have expanded by leaps and bounds. There is a demand for the preservation of fertility among transgender patients, although access may be hindered by sociological, economic, and legal barriers.

For male-at-birth patients, sperm freezing remains the only viable option. While freezing does inevitably lead to some degradation in quality, this method nevertheless has been highly successful in achieving conception. However, there is currently a dearth of research regarding the long-term impact of transgender hormone therapy on sperm quality, and even less information is available regarding live births using such sperm.

Although some fertility centers will now harvest testicular tissue from prepubertal patients for fertility preservation, techniques using such tissue are not yet feasible in humans.

For female-at-birth patients, mature oocyte harvesting and cryopreservation remain the most promising options, as it is an established method with live birth rates paralleling those of fresh oocytes. It is possible that in the future ovarian tissue preservation may represent the most suitable option, particularly for young patients, as it does not require mature oocytes and can be completed without significantly delaying testosterone treatment. However, more research is needed to determine optimal techniques for tissue harvesting and preservation, and rates of successful pregnancy vary highly by facility, suggesting the need for the development of standardized protocols.

Although limited data currently exists regarding the success rates of fertility preservation in transgender patients, parallels with other uses of ART and the success rates thereof show that fertility preservation can be a viable option. Therefore, it is important that patients be counseled on their fertility preservation options prior to undergoing medical transition.

**Author Contributions:** Conceptualization, T.A.S., D.A.O. and N.M.; methodology, N.M., P.S. and T.A.S.; data curation, N.A., A.R.A., N.M. and T.A.S.; writing—original draft preparation, T.A.S., N.M., N.A., A.R.A.; writing—review and editing, N.A., A.R.A., N.M., D.A.O., A.C., A.B., P.S. and T.A.S.; supervision, T.A.S.; project administration, T.A.S. and P.S. All authors have read and agreed to the published version of the manuscript.

**Funding:** This work was supported and funded by the Department of Urology, University of Toledo.

**Institutional Review Board Statement:** Not applicable.

**Informed Consent Statement:** Not applicable.

**Conflicts of Interest:** The authors declare no conflict of interest.

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
