# Peer review of "Fertility Preservation Options for Transgender Patients: An Overview"

_2673-4397, doi:10.3390/uro3040024_

Round 1

Reviewer 1 Report

Well written manuscript with some major lacks:

- the quotes used by this overview are dated, the 20% of the articles cited are since 2021 or recent.

- another paragraph about the future perspectives is needed; the authors might describe which techniques are more promising and how they can impact this field of research

- the conclusion is too focused on the access to the preservation and less on the research and on the techniques.

Author Response

Good afternoon!

Thank you for your review of this manuscript. You raised excellent points regarding the content, and your input was quite helpful. We agreed with all of it. The content has been revised with special attention to the following:

  • The quotes used by this overview are dated, the 20% of the articles cited are since 2021 or recent.
    • Content has been updated with multiple additional sources from 2021-2023.
  • Another paragraph about the future perspectives is needed; the authors might describe which techniques are more promising and how they can impact this field of research
    • The section on techniques has been updated to give an overview of recent research and technology in development, as well as which new techniques may provide the most patient benefit. 
  • The conclusion is too focused on the access to the preservation and less on the research and on the techniques.
    • The conclusion section has been expanded to include a summary of how current techniques differ from those in development, the potential implications of new research and techniques on the field, and possible directions for further research. 

Changes in the updated manuscript are highlighted in yellow. Thank you again for taking the time to review this!

Reviewer 2 Report

Dear Authors, I read your paper with interest, since this topic is nowadays an important issue. The revision is well written and organized. I have nothing to add. 

Author Response

Good afternoon!

Thank you for reviewing this manuscript. Your rating form mentioned that the manuscript could be more comprehensive and include more scientific background, and we agree with this feedback. The manuscript has been updated, with the following changes:

  • Content has been updated with multiple additional sources from 2021-2023.
  • The section on techniques has been updated to give an overview of recent research and technology in development, as well as which new techniques may provide the most patient benefit. 
  • The conclusion section has been expanded to include a summary of how current techniques differ from those in development, the potential implications of new research and techniques on the field, and possible directions for further research. 

These changes should provide a more comprehensive and up-to-date view of the topic. Changes in the updated file are highlighted in yellow. Thank you again for taking the time to review this!

Round 2

Reviewer 1 Report

Authors have modified the draft with an update of the bibliography.

Right now the manuscript seems to underline more the techniques with a good overview on various options.